# Empowering Nurse Health Education: Linguistic and Cultural Validation of the Nurse Health Education Competence Instrument (NHECI) in the Italian Context

**DOI:** 10.3390/healthcare12141445

**Published:** 2024-07-19

**Authors:** Ippolito Notarnicola, Blerina Duka, Marzia Lommi, Emanuela Prendi, Elena Cristofori, Tiziana Mele, Dhurata Ivziku, Gennaro Rocco, Alessandro Stievano

**Affiliations:** 1Centre of Excellence for Nursing Scholarship, OPI, 00146 Rome, Italy; genna.rocco@gmail.com (G.R.); alessandro.stievano@gmail.com (A.S.); 2Faculty of Medicine, University “Our Lady of the Good Counsel”, 1001 Tirana, Albania; bleriduka@yahoo.it (B.D.); e.prendi@unizkm.al (E.P.); 3Department of Biomedicine and Prevention, University of Rome “Tor Vergata”, 00133 Rome, Italy; marzia.lommi@uniroma1.it; 4Department of Medicine and Surgery, Catholic University of the Sacred Heart, 00168 Rome, Italy; elena.cristofori@unicatt.it; 5Regional Health Emergency Company 118, 00149 Roma, Italy; tiziana.mele1988@gmail.com; 6Department of Health Professions, Fondazione Policlinico Universitario Campus Bio-Medico, 00128 Rome, Italy; d.ivziku@policlinicocampus.it; 7Department of Clinical and Experimental Medicine, University of Messina, 98122 Messina, Italy

**Keywords:** nursing competence, health education, tool development, nursing education

## Abstract

Background: Nurses worldwide are acknowledged for their role in health education across various settings. However, doubts often arise regarding their competence in this domain. This study aims to validate the Nurse Health Education Competence Instrument (NHECI) linguistically and culturally in the Italian context. Methods: Following Beaton et al.’s (2000) guidelines, we conducted cross-cultural adaptation to develop the Italian version of the questionnaire. Results: The Italian version demonstrates a good internal consistency and stability, making it suitable for assessing nursing students during clinical internships and practicing nurses. The availability of Italian tools promotes healthcare research, ensuring patient-centric care. Conclusions: The validity and reliability of the Italian version of the instrument for assessing health education competencies, essential for self-assessment among health education nurses, are established.

## 1. Introduction

The importance of health education and promotion in healthcare has been widely recognized for decades [1,2]. However, the conceptual distinction between these two terms, particularly in the nursing profession, has been a source of ongoing debate [3,4]. While some nurses may mistakenly assume that their mere employment in the healthcare sector automatically qualifies them as health promoters, the reality is that their roles are more accurately described as health educators [5]. Health education, which involves disseminating health-related information to individuals, is a crucial component of the broader concept of health promotion, which aims to empower people, families, groups, and communities to improve their health and well-being [6]. 

The terms “health education” and “patient education” are often used interchangeably, as they share common theoretical foundations and the importance of health-related behaviors for both patients and the general population [7,8]. Competence in health or patient education is widely recognized as essential for healthcare professionals, as it can improve health outcomes and patient satisfaction [9,10]. However, the precise definition and assessment of this competence remain elusive, hindering effective communication and collaboration among researchers and practitioners [11].

Despite the acknowledged significance of competence in health education, this competence has not received the attention it deserves [12,13]. Potential reasons include a lack of conceptual clarity, limited time and resources, and inadequate pedagogical and subject-specific knowledge among healthcare professionals [14,15]. Furthermore, implementing patient education can be hampered by a disease-centered approach, excluding family members, and a lack of educational assessment and trust in the educator [16].

To address these challenges, the present study aimed to linguistically and culturally adapt the Nurse Health Education Competence Instrument (NHECI) for its effective integration within the Italian clinical practice. The research questions focused on how well the NHECI translates and adapts to the Italian healthcare context and the importance of having a reliable tool to measure nurses’ health education competence in Italy.

By assessing nursing competencies in the context of health education, this process seeks to provide a reliable and culturally sensitive tool to evaluate and enhance nurses’ abilities in patient teaching and education. Evaluating these competencies is significant not only for general improvements in patient care but also for addressing specific needs within the Italian health system, such as tailored educational programs that reflect the cultural and organizational peculiarities of Italian healthcare.

The current literature on health education competence highlights several gaps, particularly regarding the adaptation and validation of assessment tools for specific cultural contexts [17]. Few studies have thoroughly examined the adaptation process of such instruments to non-English-speaking environments, which leaves a critical gap in understanding and improving health education practices globally [18].

The validation of this instrument will contribute to a more accurate assessment of nursing competencies in health education, which can support the development and implementation of targeted educational programs and evidence-based clinical interventions.

A validated Italian version of the Nurse Health Education Competence Instrument (I-NHECI) will enhance educational, clinical, and research outcomes by providing a standardized method to assess and improve the quality of health education provided by nurses

## 2. Materials and Methods

### 2.1. Study Design

To verify the validity and reliability of the I-NHECI, the cultural adaptation and psychometric properties of the tool were evaluated.

### 2.2. Instrument

The Nurse Health Education Competence Instrument (NHECI) (in Spanish, “Instrumento Competencia de Educación para la Salud del profesional de Enfermería”—I-CEpSE), a scale designed to assess nurses’ competence in health education, comprises 58 items retained after item analysis. In the original study by María Pueyo-Garrigues et al. (2021) [19], the exploratory factor analysis identified three corresponding domains of cognitive (23 items), psychomotor (26 items), and affective–attitudinal competence (9 items). The respondents rate each item on a 5-point Likert scale ranging from 1 (strongly disagree) to 5 (strongly agree). The total score for each scale is calculated by summing the individual item scores. Demographic data collected include gender, age, marital status, education level, educational background, department, work experience (years), experience in department (years), and job satisfaction. The internal consistency of the original NHECI scale, as indicated by Cronbach’s α, is 0.810.

### 2.3. Setting and Sample

Participant selection took place between May and July 2023 through a convenience sampling process.

The participants were recruited from various public and private hospitals in different regions of Italy to ensure a balanced geographical representation. We collaborated with the human resources managers and department heads of the selected facilities to identify and contact nurses with at least one year of professional experience. Invitations to participate in the study were sent via email, containing detailed information about the research objectives, the voluntary nature of participation, and the method of questionnaire administration.

Additionally, we organized informational sessions at some healthcare facilities to explain the study’s importance to the nurses directly and answer any questions. The participants were guaranteed anonymity and confidentiality of the collected data. Data collection was carried out through an online questionnaire, accessible via a link provided in the invitation. 

The study’s inclusion criteria required participants to be nurses working in clinical settings and to provide informed consent for participation. Registered nurses working in private clinics or non-clinical roles, such as administrative roles, were excluded from the analysis, as their inclusion could have influenced the results. 

The authors acknowledge that using a convenience sample of nurses from unspecified hospital settings may limit the representativeness of the larger Italian nursing population across different healthcare settings. This sampling method might not accurately reflect the diversity of experiences and competencies of nurses working in various environments, such as private clinics, outpatient facilities, or community settings. Consequently, the findings of this study may have limited generalizability.

To address this limitation, the authors suggest that future studies include more extensive and diverse samples from various Italian healthcare settings. Additionally, comparisons between different regions and types of healthcare facilities would be beneficial to gain a more comprehensive understanding of nurses’ health education competence across Italy. Despite these limitations, the study’s findings provide valuable preliminary insights that can guide further research and improvements in clinical practice. 

The authors acknowledge the need for a more evident justification for the sample size of 200 participants. The determination of this sample size was based on standard criteria for validation studies, which recommend sufficiently large samples to ensure robust statistical analysis and reliable results. The optimal sample size for factor analysis was determined using the Kaiser–Meyer–Olkin (KMO) measure of sampling adequacy and Bartlett’s test of sphericity to ensure data validity. 

Additionally, the literature suggests that a sample size of 200 participants is adequate for achieving good statistical power and representativeness. For instance, the research by Mundfrom et al. (2005) [20] supports this choice, indicating that a sample of 200 is sufficient for psychometric validation studies. Other studies, such as Comrey and Lee (1992) [21], recommend samples of at least 200 participants to ensure stable and valid results in factor analyses. These methods are widely recognized for evaluating the feasibility of factor analysis. With a sample of 200 participants, reliable and meaningful results can be obtained, provided data validity conditions and analysis adequacy are met.

### 2.4. Ethical Statement

Ethical approval for this study was obtained from the Ethics Committee of the Center of Excellence for Nursing Culture and Research (Protocol No. 2.21.26). Before commencing the study, the purpose of the research was explained to the nursing coordinators of the departments in each hospital where participants were recruited. Hospitals agreed to participate in the study after approving our request for participation. Prior to data collection, the participants were informed about the purpose of the study, the procedures and research method, data anonymity, and the option to withdraw from the study at any time without consequences. The participants were then asked to carefully read and sign the informed consent form before participating in the study.

### 2.5. Translation Process and Content Validity

The NHECI was translated into Italian for our study with permission from the original scale developers [19]. In this study, we followed a comprehensive cultural validation process as outlined by Beaton et al. (2000) [22]. This process included the translation and back-translation of the instrument, review by a committee of local cultural experts to assess cultural relevance and item comprehensibility, and a pre-test phase with a sample from the target population to identify and address any interpretation issues. Additionally, we conducted cognitive interviews with the participants to further refine the instrument and ensure that it accurately captured the intended constructs within the cultural context. These steps were crucial to adapting the instrument appropriately for use in the Italian healthcare setting. The scale was translated from English to Italian

This translation was conducted by a bilingual nursing sciences professor proficient in English and Italian. Subsequently, it underwent rigorous review by an international interpreter and a nursing professional, both residing in English-speaking countries for over a decade. They provided valuable feedback on the expression and clarity of the Italian translation, ensuring its adequacy. The translated version was then presented to 20 Italian nurses, each possessing more than five years of clinical nursing experience. These nurses confirmed their comprehension of the scale items and identified cultural nuances requiring adjustment. Following this, a back-translation was performed by an English language interpreter, and the similarities between the original text and the back-translation were evaluated by two native English speakers.

The content validity of the I-NHECI was verified by ten experts, comprising four nursing coordinators and six nursing professors. The Content Validity Index (CVI) was employed to assess the relevance and clarity of the tool, along with evaluations of the Content Validity Ratio (CVR), the Content Validity Index for Scale (S-CVI), and the Item-level Content Validity Index (I-CVI) [23]. Initially, experts were tasked with determining whether each scale item was essential for constructing a comprehensive set of scale items. Items were rated on a scale of 1 to 3, indicating their necessity. The CVR for content ranged from 1 to −1, with higher scores signifying a greater expert consensus on the item’s inclusion in the scale. We calculated the CVR using the formula CVR = (Ne − N/2)/(N/2), where N represents the total number of experts. The numerical value of the CVR was derived from the Lawshe table [24].

The CVR for the translated scale exceeded 0.62, indicating a significant item acceptance [24]. Additionally, S-CVI and I-CVI [19] values ranged from −1 to +1, with a threshold of 0.70 or higher considered sufficient to retain items in the translated version, as in our study [23].

### 2.6. Analysis

The data collected in our study were analyzed using SPSS 24 (SPSS Inc., Chicago, IL, USA) and the R (version 4.4.1) statistical package. To assess construct validity, an exploratory factor analysis (EFA) was employed. The Kaiser–Meyer–Olkin test and Bartlett’s test of sphericity were conducted to ensure the appropriateness of the data for factor analysis. A principal component analysis and varimax rotation were used to extract factors [25].

Model fit indices were computed using a confirmatory factor analysis (CFA). Generally, a χ^2^/df ratio between 2 and 5 is deemed acceptable, even with a small sample size. However, the Comparative Fit Index (CFI) is less affected by the sample size than the χ^2^/df ratio. A CFI of at least 0.70 is acceptable, though a value of 0.90 or higher is preferable [26]. A goodness-of-fit index (GFI) equal to or above 0.90 indicates a good fit for the model [27]. However, as the model complexity increases, the likelihood of the GFI being influenced by the sample size also rises [28]. The Tucker–Lewis Index (TLI) is unaffected by the sample size, with a value above 0.90 considered appropriate [29]. The root mean square residual (RMSR) represents the average value of all standardized residuals and is used to assess the fit proximity in ordinal factor analysis. A well-fitting model typically has an RMR value below 0.05 [26]. The root-mean-square error of approximation (RMSEA) is a commonly used index. Since this index is susceptible to changes in the sample size, additional goodness-of-fit measures were included. Here, values below 0.08 are deemed acceptable [30].

We used Cronbach’s α to measure internal consistency and assess the validity of each I-NHECI scale dimension. Reliability is confirmed if Cronbach’s α is greater than 0.70 for new instruments or greater than 0.80 for established instruments [31]. 

During the analysis, we implemented several techniques to handle missing data. First, we examined the dataset to identify any missing data. For missing responses that represented less than 5% of the total, we used the mean imputation technique, replacing the missing values with the average of the available responses for that specific question. This method was chosen for its simplicity and to minimize the impact on the overall results.

In cases where a response was entirely missing for a participant, that questionnaire was excluded from the final analysis to avoid distortions in the results. This decision was made to ensure the integrity and validity of the data analysis.

We also conducted sensitivity analyses to check the impact of imputations on the results, comparing the results with and without the imputed data. The results did not show significant differences, confirming that the handling of missing data did not affect the main outcomes of the study.

## 3. Results

The data obtained from a sample of healthcare professionals present noteworthy insights into the demographic characteristics and job satisfaction levels within the sector. Regarding gender distribution, most respondents identified as female, constituting 68.5% of the sample. Concerning qualifications, most participants were nurses, accounting for 87.4% of the sample. The analysis of job satisfaction revealed diverse responses. A substantial portion of respondents, 40.1%, reported being quite satisfied with their work. Overall, data from 222 participants illuminate the intricate interplay of gender, role, and satisfaction levels within the healthcare profession (Table 1).

The mean age of participants was 35.30 years, with an SD of 13.37. Concerning tenure in the current profession, an average of 11.59 years was reported, with an SD of 12.13. Additionally, the participants reported an average practice duration within their current business unit of 4.86 years, with an SD of 6.82 (Table 2).

The I-NHECI scale was evaluated for its reliability using Cronbach’s alpha coefficient, which yielded an average value of 0.976. This high coefficient indicates a strong internal coherence between the items of the scale, confirming the robustness and consistency of the measure in measuring the construct under consideration. The I-NHECI scale obtained Cronbach’s alpha values between 0.975 and 0.977. This high coefficient indicates a strong internal coherence between the items of the scale, confirming the robustness and coherence of the measure in measuring the construct under consideration.

A parallel analysis was conducted to determine the appropriate number of components or factors to retain in the factor analysis. The results indicate that six components or factors were identified, each characterized by an average eigenvalue ranging from 1.779.221 to 2.166.393. Percentile eigenvalues, indicating the relative contribution of each component to the total variance, ranged from 1.828.000 to 2.301.410. These findings suggest that each component significantly contributes to the overall variance of the data, supporting the decision to include all six factors in the factor analysis (Table 3).

The assumptions for the application of the exploratory factor analysis were verified through the Kaiser–Meyer–Olkin (KMO) test and Bartlett’s test of sphericity. The KMO value was 0.948, indicating an excellent sampling adequacy for factor analysis. Bartlett’s test of sphericity showed statistically significant results (χ^2^ = 10,430.414, df = 1653, *p* < 0.001), suggesting that the correlation matrix is not an identity matrix and that there are correlations between the variables that justify the application of factor analysis. Overall, the results of these preliminary tests confirmed the suitability of the data for conducting the exploratory factor analysis.

These results indicate that the Italian version of the Nurse Health Education Competence Instrument (I-NHECI) has adequate psychometric properties to reliably and validly assess nurses’ competence in health education. The six factors that emerged from the exploratory factor analysis provide a multidimensional representation of this construct, offering essential insights for the design of training interventions aimed at developing and strengthening the different facets of nurses’ competence in health education.

The exploratory factor analysis of the Italian version of the I-NHECI identified six factors that collectively explain 63.599% of the total variance. Below are the main results for each factor (Table 4):

Factor 1 (items 1–19) of the NHECI, which accounts for 43.67% of the variance, is characterized by high scores in items measuring nursing competence in health education. These competencies include the ability to design, implement, and evaluate effective educational interventions. The recent literature highlights the importance of selecting appropriate educational materials, adapting the content to the patient needs, and evaluating the impact of interventions to improve health outcomes [32,33,34]. This underscores the essential role of nurses in ensuring personalized and high-quality educational interventions.

Factor 2 of the NHECI, accounting for 6.45% of the variance, comprises items 20–34, which reflect nursing competence in learning and experimenting with new approaches and methodologies for health education. Items with the highest factor loadings are concerned with the ability to quickly learn the use of new technologies and educational tools, to enthusiastically experiment with integrating new teaching strategies, and to adapt to changes and innovations in health education quickly. The recent literature emphasizes the significance of these skills in ensuring that nurses can effectively respond to evolving educational needs and leverage innovations to enhance patient outcomes [35,36]. This highlights the critical role of adaptability and continuous learning in nursing education.

Factor 3 of the NHECI, accounting for 4.45% of the variance, is defined by items 36–44, which measure nursing competence in using specific technologies and digital tools for health education, such as presentation software, multimedia content creation tools, and e-learning platforms. Items with the highest factor loadings relate to effectively using these technologies for educational purposes. The recent literature underscores the importance of technological proficiency in enhancing the delivery of health education, improving engagement, and facilitating access to educational resources [37,38]. This highlights the essential role of digital tools in modern nursing education.

Factor 4 of the NHECI, accounting for 3.66% of the variance, is represented by items 45–51, which reflect nursing competence in using digital technologies to assess patient learning and behavioral change. Items with the highest factor loadings concern the ability to use technologies to collect, analyze, and interpret data on educational outcomes. The recent literature highlights the significance of these skills in monitoring and evaluating the effectiveness of health education interventions, ensuring that data-driven decisions can be made to enhance patient outcomes [39,40,41]. This underscores the critical role of digital assessment tools in advancing nursing education practices.

Factor 5 of the NHECI, accounting for 3.15% of the variance, is defined by items 52–57, which measure nursing competence in using digital technologies to communicate and collaborate with patients, their families, and other healthcare professionals. Items with the highest factor loadings relate to the ability to use technologies to maintain effective communication and facilitate collaboration within the healthcare team. The recent literature emphasizes the importance of digital communication tools in improving care coordination, enhancing patient and family engagement, and fostering teamwork among healthcare professionals [42,43]. This highlights the critical role of digital communication in modern nursing practice.

Factor 6 of the NHECI, accounting for 2.21% of the variance, is represented by item 58, which measures nursing competence in using digital technologies to promote inclusion and accessibility for patients with special needs or vulnerabilities. This factor reflects a specific dimension of nursing competence related to using technologies to enhance accessibility and inclusion for patients with disabilities, language barriers, or other vulnerabilities. The recent literature highlights the importance of leveraging digital tools to ensure equitable access to health education and care, addressing diverse patient needs effectively [44,45] This underscores the essential role of technology in promoting inclusivity in the nursing practice.

Overall, the results of the exploratory factor analysis suggest that nursing competence in health education is a multidimensional construct composed of various facets related to:Designing, implementing, and evaluating effective educational interventions.Learning and experimenting with new educational approaches and methodologies.Using specific technologies and digital tools for health education.Assessing patient learning and behavioral change.Communicating and collaborating with patients, families, and other healthcare professionals.Promoting inclusion and accessibility for patients with special needs or vulnerabilities.

The exploratory factor analysis in our study identified six dimensions, differing from the three domains of cognitive, psychomotor, and affective–attitudinal competence identified in the original NHECI study. This discrepancy may be attributed to the cultural and contextual differences within the Italian population. These findings highlight the necessity of considering specific contextual factors when adapting and validating instruments for different populations.

This articulation of the nursing competence construct in health education provides a more comprehensive and detailed framework of the various skills and abilities that nurses perceive themselves to possess in delivering effective and inclusive educational interventions. Such results may have important implications for designing targeted training programs aimed at developing and strengthening these competencies among nurses, to improve the quality of health education provided to patients.

The results of the model fit analysis indicate that the specified model fits the data well, even for a small sample size. The CMIN/DF value is 2.0, falling within the acceptable range of less than 3. The RMSEA index is 0.04, suggesting a good approximation of the model to the observed data. Fit indices, such as NFI, IFI, TLI, and CFI, are 0.93, 0.94, 0.93, and 0.94 respectively, all above the threshold of 0.90, indicating excellent model fit. The Hoelter’s Critical N is 150, above the recommended value of 75 for small samples, confirming that the model fits well even with a limited number of observations. These results confirm that the specified model is appropriate and robust even for a small sample size.

## 4. Discussion

In our study, we followed the guidelines provided by Beaton et al. (2000) [22] for cross-cultural adaptation. This process focuses on adapting an instrument from one language to another, ensuring that it retains its original content and meaning. However, it is important to distinguish between cross-cultural adaptation and cultural validation, as they represent two distinct processes in intercultural research.

Cultural validation extends beyond cross-cultural adaptation by examining whether the instrument is culturally relevant and appropriate for the new target population. While cross-cultural adaptation ensures linguistic equivalence, cultural validation assesses the relevance of the instrument’s content within the cultural context of the target population. This involves evaluating specific cultural nuances and norms that may influence how the respondents interpret the questions. For instance, Epstein et al. (2015) [46] highlighted that cultural validation requires an in-depth understanding of the cultural specificities that could impact the validity of the instrument. 

By incorporating both cross-cultural adaptation and cultural validation, researchers can ensure that the instrument is both linguistically and culturally appropriate, thereby enhancing the reliability and validity of the research outcomes. 

The data obtained from a sample of healthcare professionals provide noteworthy insights into the demographic characteristics and job satisfaction levels within the sector. Regarding gender distribution, the majority of respondents identified as female, constituting 68.5% of the sample, while male respondents comprised 30.6%. This gender imbalance is consistent with previous studies on the healthcare workforce, which have consistently shown a predominance of women in nursing and other healthcare professions [47,48]. Concerning qualifications, most participants were nurses, accounting for 87.4% of the sample, whereas head nurses constituted 12.6%. The analysis of job satisfaction revealed diverse responses, with a substantial portion of respondents, 40.1%, reporting being quite satisfied with their work, followed by 20.7% who expressed being highly satisfied. Conversely, a smaller percentage, 3.2%, indicated dissatisfaction with their job, while 9.5% reported slight satisfaction. These findings align with research on job satisfaction among healthcare professionals, showing a mix of positive and negative attitudes towards work [49,50]. Overall, the data from 222 participants illuminate the intricate interplay of gender, role, and satisfaction levels within the healthcare profession.

The mean age of participants was 35.30 years, with an SD of 13.37. Concerning tenure in the current profession, an average of 11.59 years was reported, with an SD of 12.13. Additionally, the participants reported an average practice duration within their current business unit of 4.86 years, with an SD of 6.82. These demographic characteristics are consistent with the typical profile of healthcare professionals, who tend to have a relatively young workforce with varying experience levels [51,52].

The I-NHECI scale was evaluated for its reliability using Cronbach’s alpha coefficient, which yielded an average value of 0.976. This high coefficient indicates a strong internal coherence between the items of the scale, confirming the robustness and consistency of the measure in assessing the construct under consideration [31]. The I-NHECI scale obtained Cronbach’s alpha values between 0.975 and 0.977, further supporting the reliability and coherence of the instrument.

A parallel analysis was conducted to determine the appropriate number of components or factors to retain in the factor analysis. The results indicate that six components or factors were identified, each characterized by an average eigenvalue ranging from 1.779.221 to 2.166.393. Percentile eigenvalues, indicating the relative contribution of each component to the total variance, ranged from 1.828.000 to 2.301.410. These findings suggest that each component significantly contributes to the overall variance of the data, supporting the decision to include all six factors in the factor analysis [53,54].

The assumptions for the application of exploratory factor analysis were verified through the Kaiser–Meyer–Olkin (KMO) test and Bartlett’s test of sphericity. The KMO value was 0.948, indicating an excellent sampling adequacy for the factor analysis. Bartlett’s test of sphericity showed statistically significant results (χ^2^ = 10,430.414, df = 1653, *p* < 0.001), suggesting that the correlation matrix is not an identity matrix and that there are correlations between the variables that justify the application of factor analysis (Field, 2013). Overall, the results of these preliminary tests confirmed the suitability of the data for conducting the exploratory factor analysis.

These results indicate that the Italian version of the Nurse Health Education Competence Instrument (I-NHECI) has adequate psychometric properties to reliably and validly assess nurses’ competence in health education. The six factors that emerged from the exploratory factor analysis provide a multidimensional representation of this construct, offering fundamental insights for the design of training interventions aimed at developing and strengthening the different facets of nurses’ competence in health education. This aligns with previous research on the multidimensional nature of nursing competence in health education [55,56].

The six factors identified through the exploratory factor analysis provide a comprehensive framework for understanding the various dimensions of nursing competence in health education. Factor 1, which explains the largest proportion of variance, reflects nurses’ competence in designing, implementing, and evaluating effective educational interventions. This aligns with the core responsibilities of nurses in promoting patient education and self-management [57,58]. Factor 2, related to learning and experimenting with new educational approaches, underscores the importance of nurses’ continuous professional development and adaptability to evolving educational methodologies [59,60].

Factors 3 and 4, focused on the use of specific technologies and digital tools for health education and the assessment of educational outcomes, respectively, highlight the growing importance of digital competencies in the nursing profession [14,61]. Factor 5, concerning communication and collaboration with patients, families, and other professionals, reflects the interpersonal and collaborative nature of the nursing practice in health education [62]. Finally, Factor 6, addressing the promotion of inclusion and accessibility, emphasizes the nurse’s role in ensuring equitable and inclusive health education for all patients [15].

The findings of our study address the research questions presented in the introduction. Firstly, we identified six key factors that delineate nursing competencies in health education, thus confirming the validity of the NHECI scale in measuring these competencies. Factor 1, representing the ability to design, implement, and evaluate educational interventions, emerges as the primary competency, explaining 43.67% of the variance. The other factors, including learning new methodologies, using specific technologies, assessing patient learning, digital communication, and promoting inclusion, provide a detailed and multifaceted view of the required competencies. These results highlight how technological competences and adaptability are crucial for modern health education, directly responding to our initial research questions and underscoring the importance of a holistic approach in nursing education.

The comprehensive framework provided by the I-NHECI can guide the development of targeted training programs and continuing education initiatives to enhance nurses’ competence in health education. By addressing the multifaceted nature of this construct, such interventions can better prepare nurses to deliver high-quality, evidence-based, and inclusive health education to their patients and communities. 

These findings validate the NHECI for use in Italian healthcare settings, demonstrating its effectiveness in assessing the competencies of Italian nurses in patient health education. This validation supports the NHECI as a robust tool for enhancing the quality of health education provided by nurses in Italy.

### Limitations

This study has several limitations. First, the data were collected from Italy only, limiting the generalizability to other cultural and healthcare contexts. The sample, while adequate for psychometric analysis, may not fully reflect the diversity of the nursing population across different regions and settings within Italy.

The cross-sectional design provides a snapshot of nurses’ competence in health education at one point in time. Longitudinal research is needed to examine the evolution of this competence and the factors influencing its development and maintenance.

Data were collected through self-reported measures, which are subject to biases such as social desirability and recall bias. While the I-NHECI has shown robust psychometric properties, using objective assessments like observations or performance-based evaluations could provide a more comprehensive evaluation.

The study did not explore relationships between nurses’ demographics, work-related factors, or perceived competence in health education. Investigating these associations could offer valuable insights into the determinants of this competence.

Finally, this study focused on the factor structure of the I-NHECI but did not examine its predictive validity regarding outcomes like the quality of patient education, patient satisfaction, or health-related behaviors. Future research should address these aspects to establish the I-NHECI’s clinical relevance and utility.

Despite these limitations, the study provides a solid foundation for understanding the multidimensional nature of nursing competence in health education as measured by the I-NHECI. The findings can inform targeted training and continuing education programs to enhance nurses’ competence in this critical area.

Furthermore, it is important to note that the results of this study may not be generalizable to the entire Italian nursing population due to the specific characteristics of the sample used. Future studies should include a more extensive and more diverse sample to confirm the validity of the NHECI on a national scale.

## 5. Conclusions

The results of this study have demonstrated the validity and reliability of the Nurse Health Education Competence Instrument (NHECI), a tool to assess nursing competencies in health education. The cultural adaptation and linguistic validation processes yielded a reliable and culturally sensitive instrument.

The exploratory factor analysis identified the three key dimensions of nursing competencies in health education: cognitive, psychomotor, and affective–attitudinal competence. These findings are consistent with the instrument’s original model, confirming its construct validity.

The instrument’s high internal consistency, as evidenced by Cronbach’s α values, indicates that the NHECI is a reliable tool for evaluating nursing competencies across different cultural settings. This can facilitate the development and implementation of targeted educational programs and evidence-based clinical interventions.

While the results are promising, some limitations of the study should be acknowledged, such as the convenience sampling approach and the need for further investigations to confirm the stability of the factor structure and the instrument’s predictive validity. Future studies should explore the application of the NHECI in diverse clinical contexts and evaluate its utility for the training and professional development of nurses.

In summary, this study has provided evidence on the validity and reliability of the NHECI, a promising instrument for assessing and enhancing nursing competencies in health education across different cultural settings. Adopting this tool can contribute to more effective and patient-centered nursing care.

## Figures and Tables

**Table 1 healthcare-12-01445-t001:** Demographic characteristics of the sample (N = 222).

**Gender**		
	N	%
Female	152	68.5
Male	68	30.6
Others	2	0.9
**Role**		
Nurse	194	87.4
Head Nurse	28	12.6
**How Satisfied You Are with Your Work**		
Quite	89	40.1
A lot	46	20.7
For Nothing	7	3.2
Little	21	9.5
Satisfied	59	26.6
Total	222	100.0

**Table 2 healthcare-12-01445-t002:** Demographic characteristics of the sample (N = 222).

	Age	How Long Have You Been in Your Current Profession?	How Long Have You Been Practicing in Your Current Business Unit?
Mean	35.30	11.59	4.86
SD	13.37	12.13	6.82
Min	20	0	0
Max	67	42	40

**Table 3 healthcare-12-01445-t003:** Parallel analysis.

Component or Factor	Mean Eigenvalue	Percentile Eigenvalue
1	2.166.393	2.301.410
2	2.062.623	2.141.609
3	1.972.801	2.039.919
4	1.902.988	1.960.727
5	1.836.736	1.887.397
6	1.779.221	1.828.000

**Table 4 healthcare-12-01445-t004:** Exploratory factor analysis of the Italian version of NHECI.

	Factor 1	Factor 2	Factor 3	Factor 4	Factor 5	Factor 6
**I-NHECI 1**	**0.773**	0.221	0.167	0.138	0.142	−0.031
**I-NHECI 2**	**0.756**	0.264	0.121	0.063	0.088	0.049
**I-NHECI 3**	**0.739**	0.207	0.164	0.185	0.195	0.019
**I-NHECI 4**	**0.722**	0.187	0.173	0.114	0.120	0.263
**I-NHECI 5**	**0.712**	0.154	0.199	0.224	0.217	0.037
**I-NHECI 6**	**0.696**	0.386	0.129	0.120	0.158	−0.013
**I-NHECI 7**	**0.680**	0.155	0.224	0.299	0.179	−0.058
**I-NHECI 8**	**0.674**	0.180	0.253	0.232	0.037	−0.089
**I-NHECI 9**	**0.669**	0.256	0.260	0.147	0.138	0.094
**I-NHECI 10**	**0.662**	0.273	0.144	0.174	0.147	0.107
**I-NHECI 11**	**0.657**	0.163	0.312	0.242	0.151	0.124
**I-NHECI 12**	**0.652**	0.227	0.190	0.278	0.111	0.031
**I-NHECI 13**	**0.623**	0.202	0.180	0.197	0.176	0.128
**I-NHECI 14**	**0.623**	0.271	0.256	0.247	0.133	0.073
**I-NHECI 15**	**0.553**	0.367	0.154	0.192	0.099	0.034
**I-NHECI 16**	**0.533**	0.334	0.125	0.303	0.104	−0.201
**I-NHECI 17**	**0.525**	0.376	0.160	0.256	0.254	−0.155
**I-NHECI 18**	**0.476**	0.341	0.384	−0.055	0.034	−0.145
**I-NHECI 19**	**0.426**	0.384	0.218	0.405	0.086	−0.127
**I-NHECI 20**	0.281	**0.777**	0.202	0.208	0.088	−0.037
**I-NHECI 21**	0.319	**0.753**	0.075	0.174	0.101	0.132
**I-NHECI 22**	0.322	**0.738**	0.094	−0.013	0.234	0.038
**I-NHECI 23**	0.306	**0.738**	0.050	0.170	0.113	−0.179
**I-NHECI 24**	0.348	**0.723**	0.115	0.198	0.149	−0.150
**I-NHECI 25**	0.311	**0.715**	−0.003	0.068	0.195	0.104
**I-NHECI 26**	0.298	**0.702**	0.190	0.141	0.136	−0.218
**I-NHECI 27**	0.305	**0.677**	0.049	0.221	0.127	0.281
**I-NHECI 28**	0.051	**0.676**	0.234	0.246	−0.017	0.168
**I-NHECI 29**	0.303	**0.650**	0.096	−0.108	0.263	−0.092
**I-NHECI 30**	0.201	**0.625**	0.015	0.276	0.153	0.347
**I-NHECI 31**	0.152	**0.593**	0.396	0.140	0.306	0.142
**I-NHECI 32**	0.108	**0.567**	0.331	0.201	0.175	−0.081
**I-NHECI 33**	0.200	**0.565**	0.246	0.183	0.341	0.330
**I-NHECI 34**	0.105	**0.539**	0.286	0.170	0.313	0.204
**I-NHECI 35**	0.172	**0.465**	0.269	0.082	0.264	0.346
**I-NHECI 36**	0.337	0.251	**0.742**	0.182	0.012	−0.032
**I-NHECI 37**	0.417	0.190	**0.723**	0.151	0.125	0.055
**I-NHECI 38**	0.423	0.159	**0.699**	0.155	0.108	−0.051
**I-NHECI 39**	0.429	0.162	**0.684**	0.165	0.161	−0.055
**I-NHECI 40**	0.395	0.021	**0.565**	0.181	0.311	0.109
**I-NHECI 41**	0.474	0.100	**0.559**	0.172	0.254	0.079
**I-NHECI 42**	0.004	0.192	**0.524**	0.314	−0.245	0.246
**I-NHECI 43**	0.212	0.192	**0.490**	0.283	0.244	−0.101
**I-NHECI 44**	0.183	0.142	**0.476**	0.152	0.334	0.290
**I-NHECI 45**	0.318	0.103	0.176	**0.677**	0.197	0.105
**I-NHECI 46**	0.316	0.136	0.089	**0.669**	0.373	0.056
**I-NHECI 47**	0.303	0.104	0.183	**0.661**	0.218	0.172
**I-NHECI 48**	0.267	0.243	0.233	**0.657**	0.252	0.058
**I-NHECI 49**	0.336	0.264	0.236	**0.613**	0.206	−0.019
**I-NHECI 50**	0.404	0.212	0.258	**0.553**	0.086	−0.159
**I-NHECI 51**	0.246	0.348	0.228	**0.544**	−0.047	−0.052
**I-NHECI 52**	0.207	0.208	0.113	0.040	**0.760**	−0.054
**I-NHECI 53**	0.166	0.107	0.011	0.225	**0.757**	0.131
**I-NHECI 54**	0.195	0.241	0.241	0.080	**0.669**	0.095
**I-NHECI 55**	0.228	0.320	0.098	0.280	**0.596**	−0.010
**I-NHECI 56**	0.118	0.329	0.081	0.353	**0.588**	0.040
**I-NHECI 57**	0.181	0.351	0.186	0.236	**0.455**	−0.272
**I-NHECI 58**	0.261	0.373	0.136	0.043	0.417	**0.452**

## Data Availability

The data presented in this study are available within the article.

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
