# Peer review of "Empowering Nurse Health Education: Linguistic and Cultural Validation of the Nurse Health Education Competence Instrument (NHECI) in the Italian Context"

_healthcare, 2024, doi:10.3390/healthcare12141445_

Round 1

Reviewer 1 Report

Comments and Suggestions for Authors

The manuscript is well written and effectively communicates the findings of an important study on the validation of the Nurse Health Education Competence Instrument for the Italian context.

I appreciate if the authors consider the followings:

In the Introduction, the authors may add more context regarding the specific challenges and needs of the Italian healthcare system that make this validation particularly important.

The paragraph (lines 53-60) describes the objectives of the study but could benefit from clearer articulation and inclusion of research questions to enhance focus. E.g. What is the reliability and validity of the Italian version of the NHECI in assessing nursing competencies in health education? How can the validated NHECI support the development and implementation of targeted educational programs and clinical interventions in Italy? What are the specific challenges and considerations in adapting health education assessment tools for the Italian nursing context?

Please cite the term “culturally adapt” (line 54)

Please provide full text for the abbreviated I-NHECI (line 63).

Instrument (lines 66-75): it is not clear if the authors use the version of the scale developed by Maria PG (2021), which was coined e I-CEpSE in Spanish as it is?If yes, please cite the original name.

It is not clear the Cronbach alpha .810 (line 75) is of some previous study (please cite), or derived from the current study, which is clearly different from the .976 in the Results (line 182). Please check.

Analysis: the results for EFA is presented in the Results. However, the results of Confirmatory Factor Analysis (CFA) such as model fit indices, CFI, GFI.etc. were not presented in the RESULTS section. Please consider include the corresponding results, or remove these (lines 135-148) from the methods.

It may not need to provide full explanation about Cronbach’ alpha (lines 149~154). It could be shortened.

Lines 156~166. It seems the text repeats exactly the data presented in Table 1. Please consider shorten or remove the text redundancy.

Line 171: please consider use “mean age” instead of “average age”. “standard deviation” needs to be followed by its abbreviation “SD”.

Please consider reducing number of decimals (e.g. SD of 13.367 could be 13.37 or 13.4). This comment is applied to the whole manuscript.

Lines 171~177. Similarly, it seems the text repeats exactly the data presented in Table 2. Please consider shorten or remove the text redundancy.

Table 2. DS probably means “SD”? Please consider “average” to be “mean”.

Table 3. It is not clear why the Cronbach’ alpha for each item of the scale, and its “average value” or “mean” are being presented here, as it presents the reliability for the whole scale or sub-scale. The corresponding text (lines 181~188) also needs to be clear from unnecessary explanation. Please consider remove this table.

The text (lines 212~248) presents the EFA results, which repeats exactly the data presented in Table 5. Please consider shorten or remove the text redundancy. Please consider shortening the text, and adding a column to the Table 5 specifying % of variance.

Similar to the above comment to the Methods, please include the results of CFA such as model fit indices, CFI, GFI.etc. to this section or remove it  (lines 135-148) from the methods.

Discussion: lines 269~295 are just descriptive of the Table 1,2,3. Please provide more references and comparison with other previous studies.

Please add some discussion on “cultural validation” with references. It could be different from “cross cultural adaptation”?

Discuss the unique aspects of the Italian healthcare system and nursing education that underscore the need for this validation study.

Please add some discussion about implications for nursing education or how this tool might influence practice and policy in Italy. Highlight how the validated NHECI can impact nursing education, practice, and policy in Italy. Discuss potential benefits for patient care and healthcare outcomes.

Limitation (lines 341~369): too long. Please shorten unnecessary explanations and make it into a paragraph.

Reviewer 2 Report

Comments and Suggestions for Authors

This study presents the validation process for instrument validation, but the cultural validation steps are not described in detail, and the main validation process of the instrument is only the results of exploratory factor analysis.

If the construct validity of the original instrument is confirmed, it is generally more appropriate to conduct a confirmatory factor analysis rather than an exploratory factor analysis.

In addition, recent instrument validation studies have included a variety of validation processes in addition to factor analysis (such as convergent validity or concurrent validity and group comparison validity), which are missing from this study.

Reviewer 3 Report

Comments and Suggestions for Authors

see as attached file.

Comments on the Quality of English Language

The manuscript requires minor editing.

Round 2

Reviewer 1 Report

Comments and Suggestions for Authors

The authors have adequately addressed all my comments, and the manuscript appears to be improved. I have no further comments.

Reviewer 2 Report

Comments and Suggestions for Authors

I'd like to thank the authors for taking my comments on board and making good revisions to the manuscript. I expect it to be a good paper.

Reviewer 3 Report

Comments and Suggestions for Authors

The above version has been successfully modified according to my comments.

Well done.